# Personal+Context navigation: combining AR and shared displays in Network Path-following

Raphaël James*    Anastasia Bezerianos*    Olivier Chapuis*    Maxime Cordeil[†]    Tim Dwyer[†]    Arnaud Prouzeau[†]

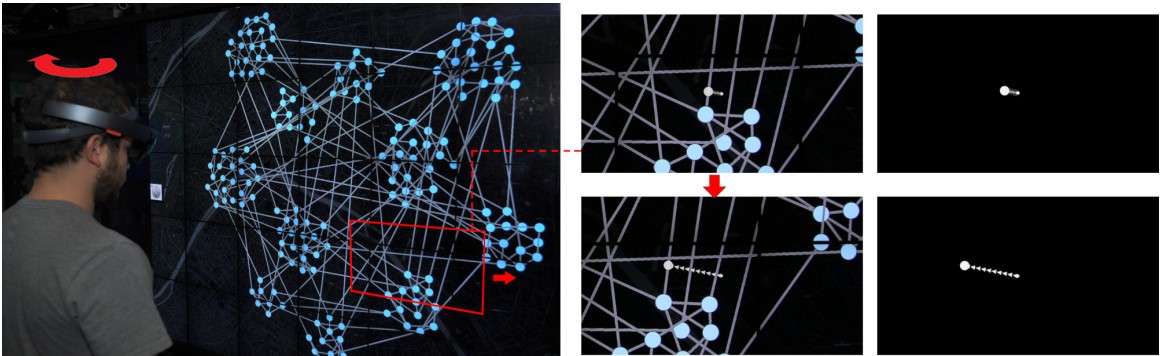

Figure 1: On the left a user with a HoloLens navigating a network shown on a shared display, moving their head from left to right. On the right their personal view in the HoloLens at the start (top) and end (bottom) of their movement. The augmented content consists only of the white visuals connecting the headset center of view (cursor) to a link of the network on the shared display (*SlidingRing* shown, red marks added for illustration).

## ABSTRACT

Shared displays are well suited to public viewing and collaboration, however they lack personal space to view private information and act without disturbing others. Combining them with Augmented Reality (AR) headsets allows interaction without altering the context on the shared display. We study a set of such interaction techniques in the context of network navigation, in particular path following, an important network analysis task. Applications abound, for example planning private trips on a network map shown on a public display. The proposed techniques allow for hands-free interaction, rendering visual aids inside the headset, in order to help the viewer maintain a connection between the AR cursor and the network that is only shown on the shared display. In two experiments on path following, we found that adding persistent connections between the AR cursor and the network on the shared display works well for high precision tasks, but more transient connections work best for lower precision tasks. More broadly, we show that combining personal AR interaction with shared displays is feasible for network navigation.

**Index Terms:** Human-centered computing—Human computer interaction(HCI)—Interaction paradigms—Mixed / augmented reality; Human-centered computing—Visualization

## 1 INTRODUCTION

Shared displays are well suited for viewing large amounts of data [4, 48, 61, 62] and for accommodating multiple users simultaneously [36, 47, 73]. Shared displays exist around us in different contexts, such as dedicated analysis environments [44], command and control centers [53, 57], and public spaces such as metro stations or airports.

While a shared display provides a common view, users often require a private view to work independently, access sensitive data, or preview information before sharing it with others. To this end,

*Université Paris-Saclay, CNRS, Inria, Orsay France.
Email: [james | anab | chapuis]@lri.fr
[†]Monash University, Melbourne Australia.
Email: [maxime.cordeil | tim.dwyer | arnaud.prouzeaup]@monash.com

existing work combines shared visualization displays with private views using dedicated devices such as desktops [23,59,66,78] mobile phones [77], tablets [40] and watches [31]. However, these separate views are typically spatially decoupled, making it difficult for users to maintain a connection between private and shared views [66].

To avoid such divided attention, recent work in immersive visual analytics [49] has combined shared displays with Augmented Reality (AR) headsets. These approaches consider the shared display and the personal AR view as tightly coupled [39, 70]: the shared screen provides the context visualization, while the AR display superimposes private information.

We investigate this combination in a specific setting. We focus on publicly shared node-link *network* visualizations, that are coupled with personal views and interaction in AR, in order to support navigation (Fig. 1). Application scenarios abound: from control rooms where individual operators [59] reroute in AR their resources using a shared traffic network map as context; to private AR route finding on a public transport network map. Fig. 2 illustrates such a scenario: multiple travelers focus on the same public information display, but may be interested in different aspects of the transportation network and do not want their personal route interests (e.g., their way home) advertised to onlookers. In these situations, AR can provide a personal view with navigational support that is tailored to the user's route priorities and preferences.

We explore two navigation techniques (with two visual variations for each), rendered only in the personal AR view, that aid the wearer in following their chosen route in the network visualization shown on a shared display. The techniques use only AR view-tracking as a means of user input, as gesture recognition and hand-held devices are not supported by all AR technologies and may be awkward to use in public settings [63]. Our hands-free interaction techniques help the viewer maintain a visual connection between their personal AR view, that may shift (for example due to small head movements), and their preferred route on the network on the shared display.

In two experiments on path following of different precision, we studied how different types of coupling mechanisms between the personal AR view and the shared network display alter user performance. Our results show that persistent coupling works well for high-precision path-following tasks, where controlling the view through the AR headset is hard; while more flexible transient coupling works

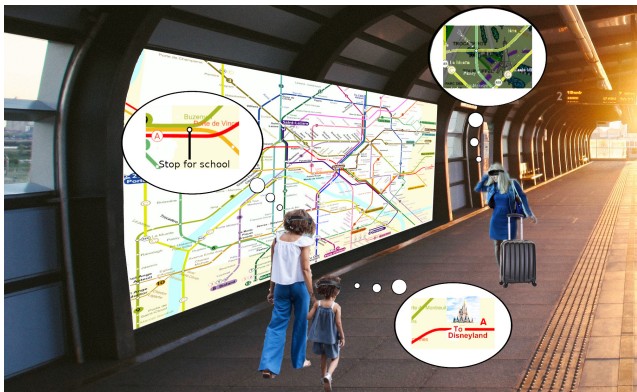

Figure 2: Personal AR route on a public map at the metro station providing geographic context. While the public map is visible to all (including people without headsets), each traveller with a headset sees their own navigational aides. The personal augmentations do not disrupt other users of the map and they give the traveller privacy.

best for low-precision tasks, in particular when following paths of (personal) high weight. More globally, they demonstrate the feasibility of having the context visualization on an external display and a personal navigation view of the AR headset.

## 2 RELATED WORK

Network and shared/collaborative visualization are well-studied topics. We focus next on the most relevant work, namely navigation and path following on network node-link diagrams (which we informally refer to as "graphs" in the sequel), the exploration of such graphs using shared displays, and the combination of AR and 2D displays.

### 2.1 Navigating Paths in Graphs

Navigating large graphs with dense links is difficult without assistance. Some solutions focus on a specific node and allow the user to explore neighboring nodes, gradually expanding their inspection of paths. Van Ham and Perer [75] use the concept of Degree of Interest to select a subgraph to present to the user and then extend it gradually. And May *et al.* [50] and Ghani *et al.* [24] use marks to highlight off-screen nodes of interest. Moscovich *et al.* [52] introduce techniques that use the topology of the graph to aid navigation, for example Bring-and-Go brings neighbors of a focus node close to it for easy selection.

All these techniques cannot work as-is in AR: although the personal view may include less elements, it still cannot override reality (the network on the shared display), adding clutter. More relevant to our work is the second technique from Moscovich *et al.* [52], Link-Sliding: after one of the neighbors is selected, the user can slide their view along a link to reach the original location of the neighbor. One of our techniques adapts this metaphor for AR settings. In our case the view of the user depends on their head orientation and cannot be forced, so instead of sliding the entire view we help the user maintain a connection to the link they are sliding on.

In a related approach, RouteLens [3] takes advantage of the topology of the graph and assists users in following links by snapping on them. Our second technique is inspired by the transient snapping [3], nevertheless we augment it with appropriate feedback to guide the users' navigation in AR.

Another family of neighborhood inspection techniques involve "lenses". EdgeLens [81], curves links inside the lens without moving the nodes in order to disambiguate node and link relationships. Local Edge Lens [74], shows in the lens only the links connected to nodes inside the lens. PushLens [65], pushes links that would transit through the lens away instead of hiding them. In MoleView [32] the lens hides links depending on specific attributes, or bundles

and unbundles them. Most of these lenses combined are seen in MultiLens [41]. Lens techniques cannot be easily applied to our context, as the video-see-through AR needs to overwrite reality, a technology not yet ready for real-world use. As with lenses, our techniques are visible in the constraint area seen through the AR display, but visuals are closely matched with the real-world content.

### 2.2 Interacting with Graphs on shared large displays

The use of shared displays for multi-user graph exploration is first demonstrated in CoCoNutrix by Isenberg *et al.* [34], where users interact using mice and keyboards. Lehman *et al.* [46] use proxemics (i.e., the distance between the user and the display) to indicate the level of zoom requested. Prouzeau *et al.* [58] compare two techniques to select elements in a graph using multi-touch. Both techniques impacted the shared workspace, potentially disturbing co-workers visually.

Lenses could limit the impact area of such interactions. For road traffic control, Schwartz [66] proposes a lens that gives additional information regarding the route on a map. Kister *et al.* [42] use the user's body position to control a lens for graph exploration. However, these lenses do not work well if two users work on the same area of the shared display, and do not address privacy issues raised in all-public workspaces. A solution is to provide users with an additional private display. Handheld devices can show detailed views [14], display labels [64], or used for interaction [40]. While such devices avoid disturbance of others and allow for privacy, they also divide the user's attention between displays [66] - the user has to match the content of their handheld with the context on the large display, which can be cognitively demanding. On the other hand, AR overlays can seamlessly combine private information with that on the shared display, avoiding divided attention. We leverage this setup to assist users in graph navigation.

Most current AR headsets provide head-tracking capabilities (e.g., HoloLens-1[1] or Nreal[2]), which is what we use in our study. Previous work has used eye-tracking as a means to analyze graph visualization and navigation [54], but not as the input mechanism for navigation. Eye-tracking is a promising alternative to hand pointing for interacting with AR content [71] and for brief interactions on public displays [38]. While some recent AR headsets do support eye-gaze tracking (like HoloLens-2[3] and MagicLeap[4], this is not yet available across the board.

### 2.3 Combining Augmented Reality and 2D displays

Combining AR with 2D displays is not new. Feiner and Shamash [21] use it to increase the size of regular displays, and Normand and McGuffin [55] to increase the size of smartphone displays. This prior work does not consider interaction.

In other works, a 2D display is used to augment AR with input that is less tiring than the mid-air gestures that are supported by the latest AR headsets. Benko *et al.* use multi-touch gestures on a tabletop to manipulate 3D objects in AR [8] and perform selections in 3D [9]. Similarly, Butscher *et al.* [13] use multi-touch gestures on a tabletop to control AR visualizations in ART. In DualCAD [51], Millette and McGuffin use a smartphone to provide 6 Degree of Freedom gestures and multi-touch interactions. This work considers external displays only as input.

Recently, AR was used to augment (add visual information to) large shared displays. Kim *et al.* [39] use AR to make static bar charts and scatterplots interactive in VisAR. Benko *et al.* [10] use it to provide a high-resolution visualization in the field of view of users, with a projector providing a low-resolution view in the

---

[1] https://docs.microsoft.com/en-us/hololens/hololens1-hardware
[2] https://www.nreal.ai/
[3] https://www.microsoft.com/fr-fr/hololens/hardware
[4] https://www.magicleap.com/en-us/magic-leap-1

periphery. Hamasaki *et al.* [28] use it in collaborative contexts to render a complex 3D scene on a large display: projectors render the view-independent components of the scene (e.g., objects, materials), and AR the view-dependent components (e.g., shadows). A similar setup is used by Zhou *et al.* [83] but in an industrial setting, with projectors rendering public information and AR private information. Our own work focuses on a different visualization context. It uses a shared large display to visualize a *network* (which is a public, view independent component), and AR headsets to display aids for personal navigation (which are private and view dependent).

In a setup similar to ours, Sun *et al.* [70] observed how collaborators communicated and interacted with each other when viewing sensitive information in an AR headset, and shared information in the wall display, in the form of an insight graph. Our work is complementary: we use a similar setup, but focus in particular in providing navigation support in the private view, that is hands-free (using view tracking in the AR headset). We also conduct a controlled study to identify the benefits of the different navigation technique designs.

Novel technological approaches, such as Parallel Reality Displays [18] can render different content depending on the viewer's position (without special headsets), thus presenting an interesting alternative for adapting and personalizing viewing content directly on the shared display. Nevertheless, they raise privacy concerns and can still not provide interaction support.

## 3 DESIGN GOALS

For the remaining of our paper, we use the term "shared context" to talk about the visualization displayed on the shared external display and "personal view" for the AR view shown to the user.

A key aspect of network navigation [30, 68, 76] is following paths of interest, while exploring their neighborhoods. Our techniques focus on aiding path following using an AR headset, while maintaining a visual link to their context (neighboring nodes and links) that exists in the network on a shared display. This gives rise to our first design goal:

*G1. Tying tightly personal view to context.* Contrary to situations where the AR view is related loosely to other displays or is a stand-alone visualization [5, 7], our techniques need to be tightly coupled to the network on the shared display. This is similar to the need to tie labels to scatterplot points in previous work [39] or to visually link virtual elements to real surfaces [60]. For graph navigation, this means the techniques need to match closely the visual structure of the underlying network on the shared display. For example any additional information on a specific node or link rendered in AR, needs to be visually connected to the shared display representation of that link or node. In our designs we vary this connection, with both *persistent* and *transient* connection variations.

We opted for hands-free techniques, as in public settings hand gestures can be awkward [63] or reveal private information (e.g., final stop on the metro map). Or they may be headset-specific or even not possible with some technologies, e.g., [20]. On the contrary, any setup that assumes a coupling of a public and private view requires view-tracking technology, that we can leverage for interaction. Nevertheless, view-tracking relies on head movement that can be inaccurate and hard to control, a situation that can be exasperated by the limited field of view of some headsets. This inspired our second design goal:

*G2. Handling limited field of view and accidental head movement.* The limited field of view of current AR headsets compounds the challenge of tying the external context display, that is often large, with the personal AR view that can cover only part of the shared display (that shows the network). This can hinder navigation, for example it is easy to get lost when following a link that starts and ends outside the field of view (especially in dense graphs). Unintended head movements aggravate this issue, as they can easily change the field of view. Thus techniques need to be robust to head movements

and changes in the field of view, reinforcing the visual connection between shared and personal view. We vary how this connection is reinforced, either providing *simple visual links* to the shared view, or a *deformed re-rendering* of a small part of the shared view to render visible information that may be outside the field of view.

We note that existing AR technology superimposes a semi-transparent overlay on the real world. As such, it is still technically hard to "overwrite" the view of the real world – in our case the shared display. Thus, the additional visual aids we bring to the personal view cannot consist of independent renderings, nor of a large amount of additional information, as the underlying visualization will continue to be visible. We thus avoided variations that require rendering large quantities of content in the field of view (e.g., remote nodes [74] or off-screen content [24, 50]), or that contradict the shared display (e.g., curving links [81] or removing them [32,65,74] since they will still be visible on the shared display).

### 3.1 Personal Navigation

When considering the motivation of having personal views on a network, we notice that path following priorities may differ depending on users' expertise and preferences. For example when looking at a public metro map, different travelers have personal route preferences (different destinations, speed, scenic routes, etc.). Thus one aspect that can be personal in network navigation is the notion of *weights* that different paths can have, depending on viewer preferences. These weights can be indicated *visually* in the AR headset (in our case as link widths), and can be taken into account in the *interaction*. Our techniques to aid path following and navigation, take into account personal path weights.

We expect that users have added their general preferences (scenic routes, high bandwidth, close to hospitals) in an initial setup phase of the system (e.g., when using it for the first time). Our techniques do not require the user to input specific/intended paths, only general preferences that many possible paths in the network can fulfill. It is then up to the user to choose and follow a specific/intended path, including ones that do not follow their general preferences. Thus our techniques do not assign weights, but make use of them.

## 4 TECHNIQUES

Our designs work without external devices, other than the head mounted display. In our setup the center of the personal view (referred to as *gaze-cursor*) acts as a virtual cursor. To navigate the graph the viewer moves their head, and thus their personal field of view, the way they would normally do when standing in front of a large shared display. Our designs do not alter the context graph visualization seen on the shared screen.

To facilitate the connection between the personal view and shared context view (*G1*) we explore two variations to aid path following: a *persistent* connection that snaps onto paths of the network permanently (Sliding), and a *transient* variation that attaches to the network temporarily (Magnetic). From now on, a subpath is a path made of links (or a link) connecting two nodes of degree 1 or $> 2$, where all the intermediate nodes, if any, are of degree 2, i.e., have a single ingoing and outgoing link (the degree of a node is the number of links that are incident to the node).

### 4.1 Persistent connection: *Sliding* Metaphor

In these variations, when a viewer is following a path on the network (i.e., one or more links), a ring gets attached to the subpath. When they change their field of view by moving their head, the ring remains attached to the subpath (that exists in the shared display), but slides along it, thus maintaining a persistent connection with the selected subpath (*G1*). Sliding is inspired by Link Sliding [52] developed for desktops, that moves the viewport of the user to follow the selected path. In our case the viewport is controlled by the viewer's head (that in turn controls their gaze-cursor), while the graph is anchored

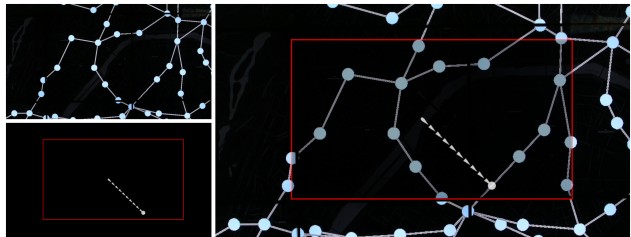

Figure 3: Example of a technique (SlidingRing) and its rendering. (Top-left) part of the shared Wall. (Bottom-left) Holographic content added by the HoloLens. (Right) Combined view seen by the user with the HoloLens on. Red border indicates the HoloLens field of view.

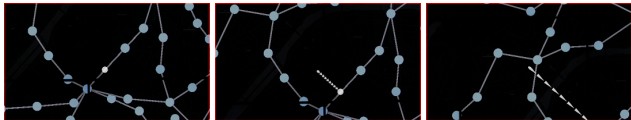

Figure 4: Close-up of SlidingRing (combined view). From left to right: as the user moves its field of view away from the link the ring is attached to, the trail (dashed line) guides the viewer back to the link, even when the ring is no longer in the AR view (right).

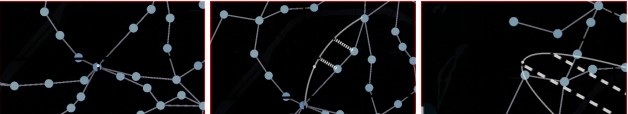

Figure 5: Close-up of SlidingElastic (combined view). From left to right: when moving away from the selected subpath, a curved copy of the path follows the user, even if the path is no longer in the HoloLens view. Again the view is the combination of the shared wall-display view (graph) and the head-mounted display (elastic path and it's connections only).

on the shared display. So we cannot force the headset viewport to follow the path. Instead, we use the metaphor of the sliding ring that gets attached to the selected subpath, and add a visual cue in the viewer's personal field of view (*G2*). This visual connection can be a simple visual link (*SlidingRing*), or a deformed re-rendering of a small part of the shared context visualization (*SlidingElastic*).

**Sliding Ring** can be seen in Fig. 3 and 4. We draw a dashed trail from the center of the personal view, to the ring that slides on a path. This ring is the projection of the gaze-cursor on the path and it slides along it as the user moves their gaze. The trail guides the viewer's eye back to their selected subpath, even if that subpath is outside their AR field of view (*G2*).

**Sliding Elastic** can be seen in Fig. 5. In this variation, the subpath in the personal view turns into a curve, pulled like an elastic towards the center of view of the user. The ring now slides on this curve. The curve can incorporate more than one node (and links). With this variation, the ring is always on the viewer's gaze-cursor (not simply attached to it via a trail), due to the deformed path curve that remains within the personal view. The end-points of the elastic curve remain tethered to the context graph (*G1*) when the viewer moves their head (*G2*), while the curve is a deformed copy of the subpath from that graph that follows the user's view. All nodes on the deformed (copied) subpath on the personal view are connected visually with dashed trails to the original nodes on the graph on the shared display. This sliding variation is more complex visually, nevertheless, as it copies the subpath, it maintains information about the local structure of the subpath in the personal view.

### 4.1.1 Interactive Behavior

For the sliding techniques we use the notion of subpaths, groups of one or more links where there are no branches and thus it is clear

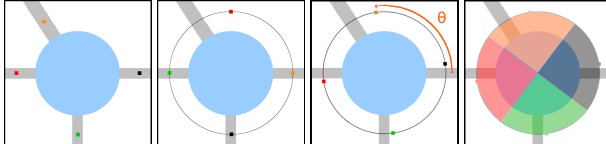

Figure 6: Choosing the link to slide into when the gaze-cursor is inside a node (cyan circle). From left to right: proxies (colored dots) are created for each link; proxies spaced equally; proxies rotated to minimize distances between proxies and original links (Procrustes analysis); slices of influence assigned to each link based on their proxies. The slice that the gaze-cursor exits on determines the link that the ring will slide to.

where the ring needs to slide to next. The ring slides on the selected subpath until it reaches the node at the end. At this point it gets attached to a new link (that may be the beginning of a longer path), depending on the direction of movement of the viewer's gaze-cursor.

At first we considered simply selecting the link closest to the center of the user's field of view. However, we found it often challenging to select between two links that are close together (like top and left link in Fig. 6). We thus provided wider zones of influence around the links, inspired by the interactive link-fanning approach of Henry Riche *et al.* [29] seen in Fig. 6. We consider proxies of all the links around the node, that are then equally spaced around a circle. The circle is then rotated in order to minimize the distance between the proxies to the original link positions (Procrustes analysis[5]). The final proxy positions define zones (slices) of influence around them. These slices follow the relative direction of the original links, but are as wide as possible to ease gaze selection. When a link has higher personal weight for the viewer, it is assigned more proxies around the circle (equal to their weight), and thus wider slices.

While this algorithm determines the choice of the next link to follow, it does not alter the visual representation of links or nodes. We observed that this approach works well even when multiple links exit towards the same direction, as viewers tend to exaggerate their head movement in a way that differentiates the links (e.g., go higher for the top link). The performance of this approach will degrade for nodes of high degree (many links) – but in these situations any technique based on head tracking, that is hard to control precisely, will be challenged.

In the case of the *SlidingRing*, the viewer's gaze-cursor can be either inside the node at the end of the subpath (as above), or further away maintaining the connection through the trail[6]. If the gaze-cursor is outside the node at the end of the subpath, we assume the viewer wants to quickly slide to the next link by roughly following with their gaze the direction of the path. We thus attach the ring to the link closer to the gaze-cursor, ignoring the fanning calculations mentioned above. This approach works well when the graph is homogeneous in terms of weights. Nevertheless, when the personal weights of the viewer are not uniform, we want to give priority to high-weight paths. These two goals (preferring the link with minimum distance and preferring one of high weight) can be conflicting, for example if a low-weight link is closer to the gaze-cursor. We assign each potential link around the node a value that thus combines their distance from the gaze-cursor and their weight. More specifically, we define two distances for each link $p_i$ with weight $w_i$. The first distance $d_{p_i}^{gc}$ is the (Euclidean) distance from the gaze-cursor ($gc$) to link $p_i$. The second distance $d_{w_i}^{w_{max}}$ is defined as $d_{w_i}^{w_{max}} = (w_{max} + 1 - w_i)$. In this term, $w_{max}$ is the highest weight in the graph, thus $d_{w_i}^{w_{max}}$ becomes smaller the higher the weight $w_i$ of the link. We refer to this distance as *distance from*

---

[5]https://en.wikipedia.org/wiki/Procrustes_analysis
[6]This does not happen in *SlidingElastic* because in the personal view the subpath curves to follow the gaze-cursor, thus the ring arrives to the node at the end of the subpath only when the gaze-cursor does.

*max weight*. The value 1 is added to our calculation of distance from max weight in order to avoid the ring always sliding to the highest weight link (where $w_i = w_{max}$), disregarding the distance of the gaze-cursor. To make our final choice of path, we combine both distances in $v_i = d_{p_i}^{gc} \times d_{w_i}^{w_{max}}$ (to be seen as the inverse of link attraction/influence) and chose the link with smallest $v_i$.

### 4.1.2 Sliding Metaphor Summary

The sliding techniques anchor the personal view to a subpath on the shared display, maintaining a consistent visual link with that path (*G1*). They are thus well suited for following a path closely during navigation. Due to their persistent coupling to paths, they are robust to changes in the field of view due to head movements (*G2*). *SlidingRing* consists of a simple dashed trail connecting the path and the viewer's field of view. *SlidingElastic* is more visually complex, providing a deformed copy of the path and its local structure, that is stretched to remains in the user's view, while the nodes at the ends remain anchored to the context graph.

## 4.2 Transient connection: *Magnet* Metaphor

In the magnet metaphor, the gaze-cursor gets "magnetically" attached to links and maintains a connection in case the user accidentally moves their head (*G2*). Only one link is attached to the magnet at a time. When the gaze-cursor moves, the attached link can change if a better candidate is found within the magnet's area of influence, thus this connection to the links on the graph of the context display is transient (*G1*). The visual connection can be again simple visual links (*MagneticArea*) or a deformed re-rendering of a small part of the shared context visualization (*MagneticElastic*).

The magnet metaphor is inspired by area cursors [37] and bubble cursors [25]. As the area of influence moves, candidate links inside the area attract the magnet. Contrary to the bubble cursor, the magnet resists detachment from the selected link to deal with small accidental head movements (*G2*), but eventually detaches in order to attach to a new candidate. The area of influence can be customized.[7] The final aspect of the magnetic techniques is providing a feed-forward mechanism to indicate risk of detachment from the current link due to nearby links. We consider attraction in an area twice the size of the area of influence to identify detachment candidates.

*Magnetic Area* is seen in Fig. 7. Similarly to the Rope Cursor [27], we draw lines from the gaze-cursor to the link that the magnet is attracted to. This simple link connects visually the gaze-cursor to the original graph on the shared display. If no links are in the area of influence, no line is drawn. The notion of a subpath is not used here, since this technique can detach from the graph. We show the attraction of other links using semi-transparent rays that fade as they move away from the gaze-cursor. The bigger the attraction the more visible the attraction rays become, acting as a feed-forward mechanism to warn the viewer that they risk detaching from the current link. To avoid visual clutter, we limit the attraction rays to the top 5 candidates.

*Magnetic Elastic* is seen in Fig. 8. Similarly to *SlidingElastic*, an elastic copy of the selected subpath is pulled into the personal view. Contrary to the sliding variation it is not permanently attached to the path and can be detached if other candidate link attract it. We communicate this attraction from other links by fading out the elastic copy when it risks getting detached from the current path and attached to another. We experimented with adding gradient attraction lines in this variation as well, but decided against them due to the visual clutter of the design. We note that for the *MagneticElastic* the notion of subpath exists, nevertheless all value calculations are done on the link of the path the viewer is on. By link we refer to the original link on the graph (and not the elastic copy).

---

[7]We set it to 5 cm on our shared display, as we empirically found this distance to be a good compromise to avoid accidental detachments when crossing over other links.

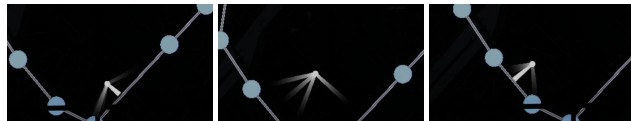

Figure 7: Close-up of MagneticArea. From left to right: the magnet is attached to a link (on the wall display); as the gaze-cursor moves other link candidates are drawn to it and attraction lines become more visible; until the magnet detaches and reattaches to another link.

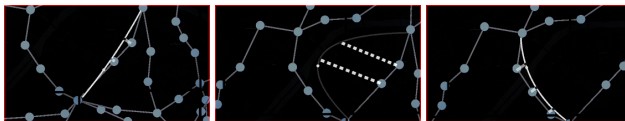

Figure 8: Close-up of MagneticElastic. From left to right: an elastic copy of the subpath is created; that follows the user even if original subpath is out of HoloLens view, nevertheless it fades out as the other links start attracting it; until it detaches and snaps to another subpath.

### 4.2.1 Interactive Behavior

When more than one link enters the area of influence of the magnet techniques, the closest to the gaze-cursor is selected. As with the sliding techniques, when the personal weights of the viewer are not uniform, we want to give priority to high weight links when considering candidates. To do so, we assign a value $v_i$ to each link $l_i$ inside the area of influence. This value takes into account both the distance of the gaze-cursor from the link $d_{l_i}^{gc}$ and the distance from max weight $d_{w_i}^{w_{max}}$ (see Sliding behaviour). Nevertheless, the simple product $v_i = d_{l_i}^{gc} \times d_{w_i}^{w_{max}}$ is not enough for techniques that are not permanently attached to links. We want a small resistance when the magnet is already attached to a link, preventing accidental detachments, especially from high-weight links.

First, we reduce detachments due to crossing. As the viewer follows a link $l_a$, their gaze may cross over another link $l_j$ resulting in a distance of zero for the crossed link. This would result in the cursor detaching from $l_a$ and attaching to $l_j$. While this detachment may be a desired behavior, we want to avoid it triggering too easily, especially for high-weight link. We thus add a term $c_1$ to our calculations of $v_i$ that prevents the distance $d_{l_i}^{gc}$ from causing immediate detachment. The new $v_i = d_{l_i}^{gc} \times d_{w_i}^{w_{max}} + c_1 \times d_{w_i}^{w_{max}} = (d_{l_i}^{gc} + c_1) \times d_{w_i}^{w_{max}}$ makes high-weight links more attractive when searching for the smallest value $v_i$. This is how values are calculated for candidate links inside the magnetic cursor's area of influence.

Second, to give priority to any currently attached link (making it resistant to detachments), we make a special calculation for the attached link $l_a$. We introduce the term $c_a < 1$ that further reduces the value of the currently attached link. The final calculation for the attached link is $v_a = (d_{l_a}^{gc} + c_1 * c_a) \times d_{w_a}^{w_{max}}$. The term $c_1 * c_a$ is always smaller than $c_1$ ($c_a < 1$), thus reducing $v_a$ with respect to the values of other candidate links inside the area of influence. We found $c_1 = 0.1$ and $c_a = 0.75$ worked well for our setup.

### 4.2.2 Magnet Metaphor Summary

The magnetic variations temporarily attach the gaze-cursor to a subpath of the graph on the shared display. The attachment prevents the user from accidentally loosing the path because of small changes in the field of view (*G2*). AR visuals are again closely tied to the context graph on the shared display (*G1*). *MagneticArea* consists of a line connecting the gaze-cursor to the selected link and rays that fade out to provide feed-forward when the viewer risks detachment. *MagneticElastic* provides an elastic copy of the path that fades when there is risk of detachment. Contrary to the sliding techniques, the link between the gaze-cursor and subpath can be broken if the gaze-cursor moves away from the path.

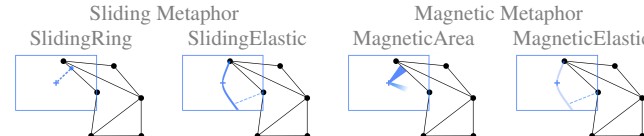

Figure 9: Schematic representations of the two interactive metaphors and their visual variations tested in our experiments (the BaseLine techniques is not shown). Visuals in **blue** indicate content rendered inside the AR headset and **black** visuals indicate content on the shared display.

## 5 EXPERIMENTAL SETUP

As the external large display we used a $5.91 \times 1.96$ m wide wall-display with a resolution of $14,400 \times 4,800$ pixels (60 ppi), composed of LCD displays (with 3 mm bezels) and driven by 10 workstations. To avoid participant movement, we only used the central part of the wall, about $2 \times 1.96$ m. For the AR part we used a HoloLens, an optical see-through head mounted display. On the wall-display we used a simple program to display images on demand and on the HoloLens side we used Unity [72]. Both are controlled by a "master" program that sends UDP messages to the wall-display (e.g., change the graph to display), and to the HoloLens (e.g., change the technique and specify the path to be followed).

We calibrated the HoloLens with the wall-display, using 3 Vuforia [33] markers rendered on the wall. The markers are recognized by the HoloLens and used to calculate an internal place-holder for the graph (position, orientation and scale). Once calibrated, the gaze-cursor of the HoloLens is projected on the wall to calculate the distances used by the techniques.

## 6 EXPERIMENTS

Our four techniques *MagneticArea*, *MagneticElastic*, *SlidingRing*, and *SlidingElastic*, Fig. 9, provide personal path navigation on a graph on an external display (*G1*), in particular under situations where the personal field of view may change due to head movement (*G2*). We chose to evaluate them under *path following tasks* [45], that are central in graph navigation and motivated their design. As the techniques aid path following (rather than path choosing), we focus on their *motor* differences, i.e., how differences in persistence and linking to the shared context visualization affect path following performance. We thus removed noise related to path choosing, by explicitly indicating to users the ideal path to follow. We also consider as a *BaseLine* the simplest path following technique: the gaze-cursor displayed in the center of the field of view of the headset. We note that the *BaseLine* is indirectly affected by path weights since higher-weight paths are rendered as wider in the headset.

Path Following Tasks. Our techniques are also designed to support different precision – the sliding techniques are very precise once on a subpath (as it is impossible to lose it), while magnetic ones are flexible and allow for quick corrections through detachments. We thus tested our techniques on path following tasks of various precision. To better understand the weak and strong points of our techniques, we decided to consider two extremes in path following w.r.t. task precision:

*1. Path Selection*: participants had to go through all the nodes and links of the path in a given order. A simple attachment (or touch/hover with the gaze-cursor for *BaseLine*) of nodes and links is enough to consider that part of the path selected. This is a low precision task that consists of a sequence of simple selections that a user may want to perform when identifying a path of interest, without necessarily being interested in all the details of the path. An example use-case could be to quickly plan a trip on a metro map or a bus network.

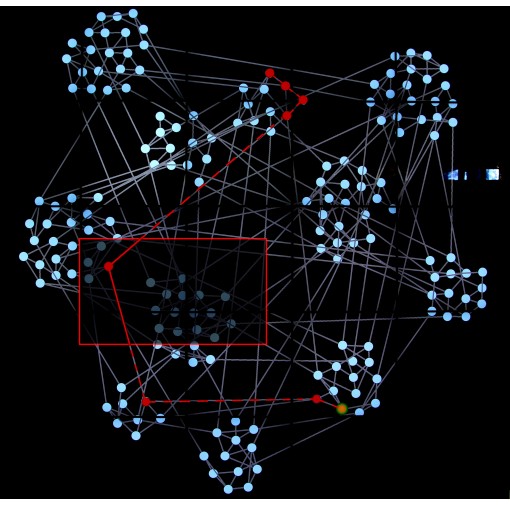

Figure 10: Example of one path from the experiment, on the Small World graph. Inside the red rectangle is the user's view through the HoloLens, that includes the red path they need to follow. The rendering outside the HoloLens view (dashed lines) are added for illustration only.

*2. Path Tracing*: participants traverse the entire path in detail, tracing nodes and the entire length of the links. If the user leaves a link, they have to return to the link and resume where they left off, until they trace the entire link. This is a high precision task that a user may perform if they are interested in details along the path they are following. An example use-case may include tracing a metro path on a geographical map to look at the specific areas it goes through, or following a road to check which parts have emergency stops or bike lanes. In real life, examples of path tracing tasks are common when networks are part of a geographic map (e.g., roads, infrastructure maps), where users are interested both in the details of the points of interest and in their context, as discussed by Alvina et al. [3] (work that partially motivated our magnetic behavior). Beyond maps, path tracing tasks are used in network evaluations when there are concerns that the visual continuity of paths in the network may be affected, for example when they cross bezels [17, 19] or curved screens [67]. In our case, visual continuity of the path may be affected by the limited field of view of the AR display (design goal *G2*).

Together, these two tasks represent extreme cases in terms of the need for interaction precision when going through a route: simple selection vs. tracing/steering along the path.

*Task Operationalization.* In real use, our techniques do not know the exact path a user would like to follow, they only know *global preferences* of the user such as preference for scenic routes or speed (that are represented as weights, see Section "Personal Navigation"). To test hypotheses in a controlled experiment, we need to isolate/control aspects relevant to the hypotheses (operationalization of the task). In our case, we have formed working hypotheses around navigation performance under different precision tasks, which we will formalize later in this section. We thus chose to remove factors that make trials incomparable, such as personal biases, preferences and decision making. In real-life, navigation requires decisions influenced by participant preferences, and thinking delays. To remove these confounds, we decided to show participants a specific route to follow in a given color (red). This ensured that only interaction time was measured (not the time to think while choosing); and avoided participants from selecting routes of different lengths based on personal preferences that could make trials incomparable across participants.

In our experiments, all experimental indications were shown in the personal AR view. As mentioned above, for experimental proposes, the path to follow was shown in red, with the starting

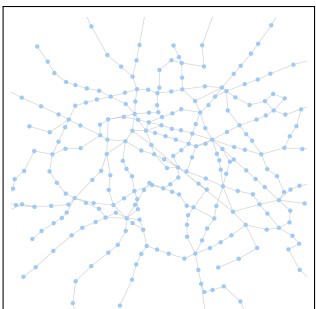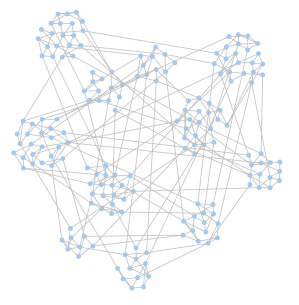

Figure 11: Graphs used in the experiments: (Left) The Quasi-Planar graph of the Paris metro map, 302 nodes and 369 links. (Right) the small world graph, 180 nodes and 360 links.

node highlighted with a green halo (Fig. 10). In the Path Selection task, whenever the participant selected a node or link in the correct order, they would turn green. In the Path Tracing task, the links were progressively filled in green up to the point the participant had reached, tracing their progress.

We separated our evaluation into two experiments, one for each of the two tasks (Path-Following, Path-Tracing).

Paths and Weights.    As we discussed in the Personal Navigation section, link weights are an example of personal information that may be specific to different users. As such, weights are a fundamental aspect of the techniques we introduced. The techniques favor high-weight links, and thus should perform better when following paths with high weights. To evaluate this aspect we considered two types of paths (factor PATH). We note that in real use, multiple paths can fulfill user preferences/constraints and could thus have high weights. For operationalization purposes we test two cases. (i) We tested *Weighted* paths where the links of the path to follow have a constant weight of 3 and all the other links have weight of 1. These simulate cases where the path that the system has identified as having high weight (based on a-priori preferences), matches the path that the user wants to follow in practice. (ii) We also tested *Homogeneous* paths where all the links of the graph have a weight of 1. With these paths of equal weight, we simulate cases where the system has assigned multiple paths with the same weight based on a-priori preferences. If the weights selected by the system do not match the needs of the user, we assume the user will deactivate them (leading them to the equal weight situation). Thus we did not consider *Homogeneous* paths that cross *Weighted* paths (an unfavorable situation for the techniques), as in real applications we expect that it would be possible to simply disable the weights.

In the experiment, links are visually scaled by their weights, hence a high weight link is 3 times wider than other links.

Graphs.    To be able to generalize our results, we considered two graphs with different characteristics (Fig. 11). The first type is a "5-quasi-planar" graph[8] (the Paris metro map). None of the paths participants had to follow contained crossings for this type of graph. The second type is a small world graph generated using NetworkX[9]. We generated the small world graph to have on purpose a similar number of links to the quasi-planar graph (369 and 360 respectively), but a higher link density. Link density is defined as *numberOfLinks/numberOfNodes*. In the small world graph this ratio was 2, almost double that of the quasi-planar graph (1.2). Consequently, the quasi-planar graph has more nodes than the small

---

[8] A topological graph is k-quasi-planar if no k of its edges are pairwise crossing [69].

[9] NetworkX (https://networkx.github.io/): we used the connected_watts_strogatz generator for the network structure, and the yEd Organic Layout.

---

world graph (302 and 180 respectively). We tested graphs of different density, as this may affect our techniques (e.g., cause accidental detachment in the magnetic techniques). The quasi-planar graph is a typical example of networks such as roads, electricity, or water networks. Small world graphs are typical in real phenomena, e.g., social networks.

The small world graph is more complex than the quasi-planar graph, because it has more link crossings, more links attached to a node on average (higher degree), and also has longer links (e.g., links between communities) that are more challenging to trace. Our aim is to see whether the structure of the graphs impacts the differences between the techniques.

In our experimental trials, all paths that participants had to follow consisted of 7 links (and 8 nodes) of similar difficulty for each graph and did not contain cycles. All the paths we used for the small world graph had one long link. An example path is seen in Fig. 10.

Working hypotheses.    Given the design of the techniques and the nature of the tasks, we made the following hypotheses:

*(H1)* For the selection task, magnetic variations are more efficient for *Homogeneous* tasks, since they do not force users to follow the entire path when trying to make simple selections.

*(H2)* For the tracing task sliding variations are more efficient than magnetic since the user needs to trace the path without detaching. Magnetic ones are likely more efficient than *BaseLine* as they can help keep the connection to the path, especially for *Weighted* paths.

*(H3)* All the techniques are faster with the *Weighted* paths than with the *Homogeneous* paths. The differences are more pronounced for the small world graph than for the quasi-planar graph, since the larger density may cause detachments in magnetic variations and distraction in *BaseLine*.

### 6.1  Path Selection Experiment

In this experiment participants conducted a low precision path following task, selecting in order links and nodes in a path.

Participants and Apparatus.    We recruited 10 participants (8 women, 2 men), aged 25 to 42 (average 29.6, median 27.5), with normal or corrected-to-normal vision[10]. Five participants had experience using an AR device, such as the HoloLens. Participants were HCI researchers, engineers, or graduate students. As apparatus, we used the prototype described above.

Procedure and Design.    The experiment is a [5×2×2] within-participants design with factors: (i) TECH: 5 techniques: *BaseLine*, *MagneticArea*, *MagneticElastic*, *SlidingRing*, *SlidingElastic*; (ii) PATH, 2 types of paths: *Weighted*, *Homogeneous*; (iii) GRAPH, 2 types of graphs: *Q-Planar*, *SmallWorld*.

We blocked trials by TECH, and then, by PATH. We counterbalanced TECH order using a Latin square. We also counter balanced PATH, for each order one participant started with *Weighted* and another with *Homogeneous*. We fixed the graph presentation order, showing the simpler *Q-Planar* first.

For each TECH × PATH, participants started with 6 training trials, followed by 6 measured trials. After each TECH block they rested while the operator checked the HoloLens calibration.

Participants were positioned 2m from the wall and were instructed to avoid walking, to maintain a consistent distance from the wall and personal field of view, across techniques and participants. They were asked to perform the task as fast as possible. Our main measure is the time to complete the task, since paths need to be completed for the trial to end (i.e., all trials are successful). Time started when participants selected the starting node, and ended when all nodes and links were selected in the correct order. The experiment lasted around 1 hour, concluding with participants ranking the techniques and answered questions on fatigue and perceived performance.

---

[10] None of our participants had red-green colorblindness, but if replicating this work other experimental colors can be considered [79].

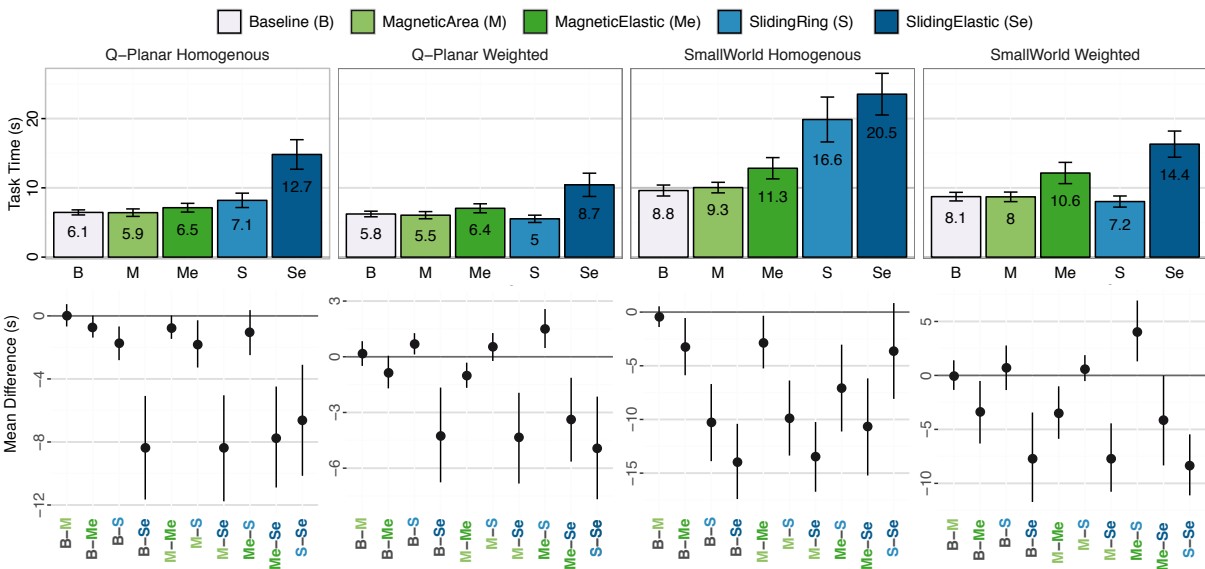

Figure 12: (Top) Mean time to complete the path selection task per TECH for each GRAPH × PATH condition (the number in the bars is the mean, and the error bars show the 95% CI for the mean over all data points). The lower the bar, the faster the technique. (Bottom) The 95% CI for mean differences between all the TECH, for the respective GRAPH × PATH condition. *Note: A CI that does not cross 0 shows evidence of a difference between the two techniques - the further away from 0 and the tighter the CI, the stronger the evidence. Not crossing 0 indicates a "significant" difference in the corresponding paired t-test, i.e., $p < 0.05$. To compare two TECHs X and Y for a given GRAPH × PATH, see first the two corresponding bars in the top figure for this condition, and then the CI that corresponds to the pair X − Y at the bottom.*

### 6.1.1 Results

We analyze, report, and interpret all inferential statistics using point and interval estimates [16]. We report sample means for task completion time and 95% confidence intervals (CIs), indicating the range of plausible values for the population mean. For our inferential analysis we use means of differences and their 95% confidence intervals (CIs). No corrections for multiple comparisons were performed [15, 56]. We also report subjective questionnaire responses.

**Completion time.** We removed one obvious outlier (a trial with a standardized residual of 18, while all others $< 4$). We did not find evidence for non-normal data. Fig. 12 shows the mean completion time for each TECH grouped by GRAPH × PATH (top) and the mean differences between TECH (bottom).

None of our techniques outperformed *BaseLine*. We see that *BaseLine* and *MagneticArea* exhibit, overall, the best performances with very similar mean task completion times across conditions (no evidence of difference). Both magnetic variations perform better than sliding ones when it comes to *Homogeneous* graphs, with strong evidence of this effect for the more complex *SmallWorld* graphs (partially confirming [H1]). Nevertheless, for *Weighted* the simple sliding variation *SlidingRing* performs well and even outperforms the complex elastic magnetic variation *MagneticElastic*.

We have evidence that *MagneticArea* and *SlidingRing* are always faster than their elastic versions *MagneticElastic* and *SlidingElastic* respectively. This is particularly clear for magnetic variations across the board, and for all cases except *Homogeneous SmallWorld* for sliding variations.

We observe mean times for each technique tend to be faster for the *Weighted* paths. However, we only have conclusive evidence they are indeed faster for *SlidingRing* (CI [1.4, 3.9]) and *SlidingElastic* (CI [2.0, 6.8]) for the *Q-Planar* graph. And for all the techniques except *MagneticElastic* for the *SmallWorld* graph (e.g., *BaseLine* CI [0, 1.7], *MagneticArea* CI [0.7, 1.9], *MagneticElastic* CI [−1.3, 2.7], *SlidingRing* CI [8.2, 15.5], *SlidingElastic* CI [3.2, 11.2]). This partially confirms [H3].

**Questionnaire.** Although in terms of time both *BaseLine* and *MagneticArea* performed similarly, the best rated technique is

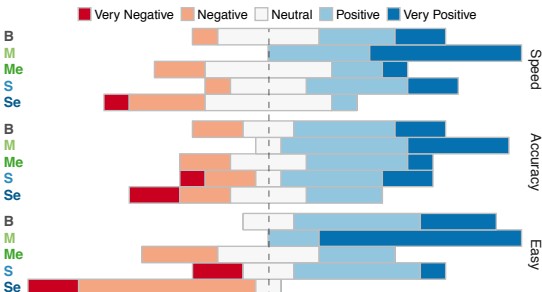

Figure 13: Results of the questionnaire of the selection experiment for perceived speed, accuracy and easiness of use of the techniques.

*MagneticArea* (average rank: 1.6). *BaseLine* is next (ar: 2.4), followed by *SlidingRing* (ar: 3.0), *MagneticElastic* (ar: 3.2), and *SlidingElastic* (ar: 4.3). Participants preferred *MagneticArea* and *SlidingRing* over their elastic counterparts.

We have very similar results when comparing responses for their perceived speed, accuracy and the easiness of use of the techniques (Fig. 13). *MagneticArea* was always better rated than the other techniques, with *BaseLine* coming second.

**Summary.** The most preferred technique is *MagneticArea*, even though it objectively performs similarly to *BaseLine* (not confirming [H1]). *SlidingRing* also exhibits good time performance for weighted paths. There is evidence that with few exceptions, techniques performed better in *Weighted* paths (partially confirming [H3]). Finally, the elastic versions performed worse than their non-elastic counterparts, possibly because the elastic variations have more visual clutter, and require users to go through the longer elastic subpath.

### 6.2 Path Tracing Experiment

In this experiment participants conducted a high precision task, tracing over each link and node in an indicated path.

Given the results of the first experiment we excluded the elastic version of the techniques. We thus consider only 3 techniques in

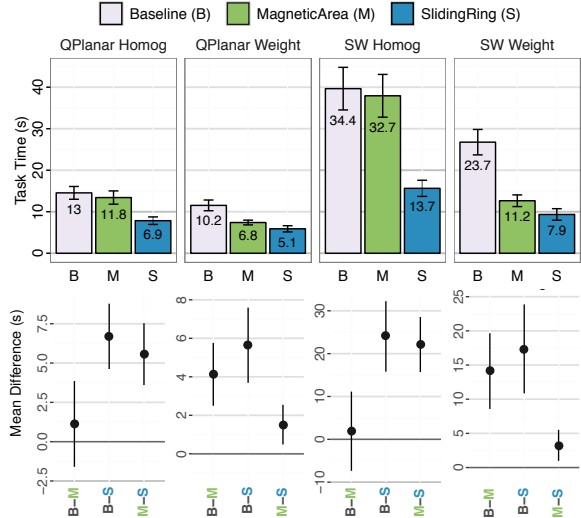

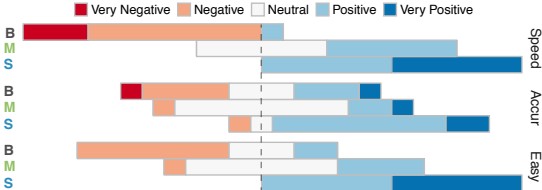

Figure 15: Results of the questionnaire of the tracing experiment for perceived speed, accuracy and easiness of use of the techniques.

Figure 14: (Top) Average time to complete the path tracing task per TECH for each GRAPH × PATH condition. (Bottom) The 95% CI for mean differences (12 points, one by participant) between all the TECH for the respective GRAPH × PATH condition. *See Fig. 12 for reading CIs.*

this second experiment: *BaseLine*, *MagneticArea*, *SlidingRing*. We follow closely the design of the first experiment and consider the same two graphs and two types of PATH.

Participants.  We recruited 12 new participants that did not participate in the first experiment (3 women, 9 men), aged 22 to 43 (median 26), with normal or corrected-to-normal vision. They were graduate students and engineers. Most had already used an AR or VR headset (five had used a HoloLens).

Procedure and Design.  We used the same procedure and design as the previous experiment, but with 3 techniques. The experiment lasted about 45 minutes.

### 6.2.1  Results

Completion time.  We removed two outliers (standardized residual $> 4$), and we could not observe any strong evidence for non-normal data. Fig. 14 shows the task completion times for the task (top) and the 95%–CI for the difference in mean for the TECH by GRAPH × PATH conditions (bottom).

We have strong evidence that *SlidingRing* is always faster than *MagneticArea* and *BaseLine*, with large differences with *BaseLine* overall, and with *MagneticArea* for *Homogeneous* paths. Moreover *MagneticArea* is clearly faster than *BaseLine* for *Weighted* paths, but this is not the case for *Homogeneous* paths. Thus, hypothesis (H2) is supported for the most part. The poorer performance of *MagneticArea* for *Homogeneous* paths is probably caused by accidental detachment when the gaze-cursor came close to a link not in the path (e.g., accidentally attaching to links that cross the path).

When comparing the performance of each TECH over the type of path, we observe that they are always faster (clear evidence) for the *Weighted* paths than for the *Homogeneous* paths (*BaseLine* for *Q-Planar* CI [1.8,4.1] and for *SmallWorld* CI [7.9,17.8], *SlidingRing* for *Q-Planar* CI [0.5,3.3] and for *SmallWorld* CI[3.5,8.9], *MagneticArea* for *Q-Planar* CI [4.0,7.9] and for *SmallWorld* CI [17.8,32.4]). This effect is particularly strong for *MagneticArea*. These results support (H3).

Questionnaire.  Overall the results of the questionnaire are consistent with the results on time. When asked to rank the techniques based on preference, 9/12 participants ranked *SlidingRing* first, one ranked *MagneticArea* first, and one ranked *BaseLine* first. *MagneticArea* was generally ranked as the second choice (for 9/12 participants). While *BaseLine* was mostly ranked last (8/12).

We observe similar results for perceived speed, accuracy and ease of use of the techniques (see Fig. 15). *SlidingRing* was judged faster, and easier to use than the other techniques. And responses for *MagneticArea* tended to be more positive than *BaseLine*.

Summary.  *SlidingRing* exhibits the best results (objective and subjective) for path tracing tasks, partially confirming [H2]. The results for *MagneticArea* depend on the nature of the path, and might be a good choice for *Weighted* paths. As expected *BaseLine* exhibits poor performance. Finally, *Weighted* tasks were faster [H3].

## 7  DISCUSSION AND PERSPECTIVES

Our results show that depending on the goal of the users, different techniques are appropriate. In an explanatory path-following task where precision may be less important, e.g., searching for a specific node or searching for the end of a link, a-priori no specific technique is needed (*BaseLine*). But viewers tend to prefer a snapping mechanism (*MagneticArea*) that is more forgiving to small view changes. When precision is important, for example when a well-determined path needs to be followed closely, a technique that is tightly coupled to the graph like *SlidingRing*, and to a lesser degree a snapping technique like *MagneticArea*, are clearly better. This is particularly true for personal paths that have higher weight, as sliding variations can follow them without risk of detaching (for instance, *SlidingRing* can be three times faster than *BaseLine*).

The elastic technique variations performed poorly. However, they can potentially be interesting when the viewer has identified a path of interest and needs to keep its local structure in their field of view. For example, in a situation where they may want to see parts of that path together with other locations on the graph.  In the future we will perform a longitudinal study to observe the long-term situational use of our techniques. As new generation AR headsets are equipped with eye-tracking capabilities, we can utilize eye-tracking analysis as a means to better understand the true focus of participants when using our techniques in practice (as has been done in the past for understanding differences in graph visualization techniques [54]).

One implication of our work is that in a real system users will need to fluidly switch between techniques depending on their goal. More generally, we expect detailed tasks will require close coupling between private and shared views, whereas in more coarse exploratory tasks this coupling can be transient. We plan to investigate this technique transition in the future.

We expect that since our techniques address accidental head-movements (*G2*), they can also be applicable in newer AR headsets that use eye-gaze instead of head-tracking (eye-tracking being considerably more noisy and prone to small movements [22, 35]). Nevertheless, this requires empirical validation.

Although path following is a common task in graphs [45], in the future we plan to explore how the techniques, and their transition, fair in complex and high-level exploration tasks in networks. For example, they may be combined with additional interactions for graph navigation and exploration, such as filtering, relayout, etc. To support a larger set of interactions we may need to consider combinations of alternative input devices (e.g., clickers or smartphones) that can provide additional input. Moreover, we expect our findings to hold for other contexts of steering-type tasks that are common in HCI literature [1], but this needs further study.

In our work we consider that viewers may have their own preferences for traversing certain paths. We model these personal link preferences with a simple weighting of each link. As the AR headset used is stereoscopic, it is tempting to use 3D to show such link weights (e.g., height above the display surface [82]). Indeed, graph visualization (such as node-link diagrams) in stereoscopic immersive environments has been proven useful under certain conditions [2, 6, 43, 80]. Nevertheless, using depth or other 3D cues to indicate weight is not straightforward in our case, where the personal view is very tightly coupled to the context graph in the external display. In our early attempts we saw that the 3D copies of subpaths with high weight may get rendered further away from their original paths on the shared display, or artificially overlap other paths. This creates a discontinuity between the personal and shared view and requires further consideration.

In light of our promising results, as technology improves more techniques could be adapted for Personal+Context navigation in node-link diagrams. Obvious candidates are magic lenses [12], fanning [29], and Bring-and-Go [52]. For instance, when over a node, the user can trigger an adapted bring-and-go that brings neighbours into the AR field of view (rather than on screen [52]). Then the user can select a neighbour, resulting in an indication in the HoloLens that points towards the direction where the selected node can be found (since the viewport cannot be forced).

Our results show that having personal views tied to a shared external visualization can aid graph navigation. The fact that these personal views are always tied to the shared display, means they can be directly applied in a multi-user context [70]. The shared graph remains visible to all users, and their personal preferences and views are private to their headset, not impeding the view of others. We next plan to investigate collaborative analysis settings, where colleagues may manipulate the graph on the shared display from their private view (e.g., scale it, move nodes, etc.). This may have several implications, such as change blindness [11] when one's AR view overlaps, but is out-of-sync, with these changes.

This raises the more general question of mismatch between reality and augmentation. Our techniques have been designed to add simple visuals, that are tightly coupled to the graph on the large display. Our decision is in part based on technological limitations of AR headsets (that can still not completely overwrite reality). Nevertheless, we feel this limited augmentation of reality, and limited movement for interactions, is appropriate in public settings (for example use in navigating public metro maps), given the recent discussions on isolation, acceptability and ethical concerns of using head-mounted displays in social settings [26].

## 8 CONCLUSION

We present two empirical studies on using personal views in augmented reality, that are tightly coupled to a shared visualization on an external display. We consider a node-link network as the shared context visualization, and use the AR view to display personal weights and to provide feedback to aid navigation. This approach could allow several users to navigate the shared graph, receiving personalized feedback, without visually cluttering the shared visualization.

Our hands-free techniques are designed to help viewers maintain their connection to the shared network visualization, even when their personal field of view changes due to small head movements. Results show that our adaptation of the link sliding technique, that is tightly coupled to the shared graph, can bring a substantial performance improvement when precisely following a path. And that a technique using a magnetic metaphor performs well and is preferred over a standard gaze-cursor for a simpler path selection task.

More globally, our work shows that Personal+Context navigation, that ties personal AR views with external shared network visualizations, is feasible; but that the nature of interaction and visual support needed to maintain this connection depends on task precision.

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
