# OpenReview forum: "Personal+Context navigation: combining AR and shared displays in Network Path-following"
_graphicsinterface.org/Graphics_Interface/2020/Conference — GI 2020_

### Official Review · AnonReviewer1 · 2020-04-20
**Interesting interaction techniques, but hard-to-follow presentation**

**Rating:** 7
**Confidence:** 3

**Review:**

This paper describes a series of overlay techniques appropriate for an augmented reality device used in concert with a shared public display, in particular for the task of graph traversal. The techniques are designed to help a user do two things: select and follow a path, and find their way back to a selected path/node when it is out of their immediate field of view. The authors describe two studies on path following tasks that make use of the system, demonstrating that for certain kinds of tasks and in particular configurations it is superior to a baseline system.

Overall, I enjoyed this paper, but it did take a couple of readthroughs to orient myself (and I am very grateful for the video figure). The words “personal” and “context” are used throughout the paper in a variety of different ways, and I found this confusing to follow (for example, on page 6 “the additional context we bring to the personal view” sort of collapses the distinction between these words for me); it would be very helpful if the authors had an explicit sentence upfront which said  “context” is the view that ... , “personal” is a view that… and used the words consistently throughout the document. In addition, the four experimental conditions tested in the user studies are never all laid out until the study design is described; in the intro the authors mention that they “explore variations of *two* navigation techniques”, when in reality it’s more of a 2x2 design? A figure demonstrating the 4 designs tested all together might be helpful? (And perhaps the authors could invert the colors in their figures to make them more legible?)

I do believe that the previous reviewers’ comments about “selection” vs “tracing” and the way in which weights were used for the user studies have been adequately addressed in the revised text.

The added (orange) text on page 7 could use a thorough proofread, the authors use “hypothesis” instead of “hypotheses”, leave unclosed parentheses, etc. I don’t know what “Procrustes analysis” is, mentioned on page 5. Another specific location where I had trouble was the ‘Questionnaire’ section on page 10 for the Path Tracing Experiment, whose first paragraph gives a lot of data in a challenging fashion but doesn’t sum up what it means. In general, I found the variety of ways the authors refer to specific components of their work challenging to keep up with, but the technical meat appears to be there.

Clarity of presentation aside, the authors have described an interesting set of techniques that seem to be quite promising! I have no qualms with the methodology of the user studies performed, and the results are definitely interesting.

I believe one piece of related work is missing: Parallel Reality Displays by Paul Dietz and Matt Lathrop, presented at SIGGRAPH ‘19. While they’re a technology rather than an interaction technique, they also provide personal/private information in public display scenarios.

---

### Official Review · AnonReviewer3 · 2020-04-21
**Review of Personal+Context navigation: combining AR and shared displays in Network Path-following**

**Rating:** 7
**Confidence:** 4

**Review:**

I appreciate that the authors of this submission produced a rebuttal and highlighted in orange the changes compared to their previous submission.

I am R3 for both submissions.

I am satisfied with the changes applied to most of the concerns that I had raised:
https://openreview.net/forum?id=VtYxuiX_UQ&noteId=3bmiwl6I6

In the new Related Work paragraph about eye-tracking, I am not sure that sentence "We thus conducted our study using head-tracking." is fair use, as we (authors and reviewers) know that this paragraph on eye-tracking has been added after the choice of using head-tracking, so the decision was not informed as it appears to reads now. I honestly think that this minor lack of clarity can be addressed.

The companion video seems identical to the first submission, based on file date.

In conclusion, I thus keep my rating unchanged.

---

### Official Review · AnonReviewer2 · 2020-04-21
**Revised submission has addressed key concerns**

**Rating:** 7
**Confidence:** 4

**Review:**

Thank you for submitting a revised version of this submission, and addressing concerns raised in the previous round of reviews.  I reviewed the previous submission as R2.
https://openreview.net/forum?id=VtYxuiX_UQ&noteId=-plgEt5-Mv

The submitted modifications show a marked improvement in the exposition of the work.  In particular, clarifications around the motivation behind the path tracing task, and additional related work that have utilized path tracing to determine endpoints (e.g., [17], [18]) and to mark or detect features along a path (e.g., [66]) were helpful in positioning the contributions of this work in relation to prior work.

I am satisfied with the changes in the modified manuscript, and changing am my recommendation to accept.

However, I noted that there are several typos throughout the text, and I recommend a thorough editing pass for the camera ready.  For example, page 3: “HoloLense” -> “Hololens”.

---

### Meta-Review · Area_Chair1 · 2020-04-23

**Recommendation:** Accept
**Confidence:** 4

**Metareview:**


This paper is a resubmission; two reviewers provided reviews on the original submission.  Both reviewers report that their concerns regarding the paper have been addressed in the revised version of the manuscript.

The reviewers agree that the paper presents a promising set of techniques to tackle topical problem for augmented reality. Based on first- and second-round reviews, the highlights of this paper include clear definition of design goals, and thorough reporting of study results.

Based on the reviewer ratings, I recommend acceptance of the paper at GI 2020, with minor corrections:
1.	Improve clarity around the description to use head-tracking instead of eye-tracking [R3]
2.	Use consistent terminology to describe concepts, techniques, and methods in the paper [R1]
3.	Proof-read added text (orange) and overall manuscript for typos [R1, R2]

---

### Decision · Program_Chairs · 2020-04-25

Accept